# Extracts and Scirpusin B from Recycled Seeds and Rinds of Passion Fruits (*Passiflora edulis* var. Tainung No. 1) Exhibit Improved Functions in Scopolamine-Induced Impaired-Memory ICR Mice

**DOI:** 10.3390/antiox12122058

**Published:** 2023-11-29

**Authors:** Yi-Yan Sie, Liang-Chieh Chen, Cai-Wei Li, Ching-Chiung Wang, Cai-Jhen Li, Der-Zen Liu, Mei-Hsien Lee, Lih-Geeng Chen, Wen-Chi Hou

**Affiliations:** 1Ph.D. Program in Clinical Drug Development of Herbal Medicine, College of Pharmacy, Taipei Medical University, Taipei 110, Taiwan; d339110003@tmu.edu.tw (Y.-Y.S.); crystal@tmu.edu.tw (C.-C.W.); lmh@tmu.edu.tw (M.-H.L.); 2School of Medicine, College of Medicine, National Sun Yat-sen University, Kaohsiung 804, Taiwan; lcchen0908@mail.nsysu.edu.tw; 3Graduate Institute of Pharmacognosy, Taipei Medical University, Taipei 110, Taiwan; licaiwei@tmu.edu.tw (C.-W.L.); m303108003@tmu.edu.tw (C.-J.L.); 4Traditional Herbal Medicine Research Center, Taipei Medical University Hospital, Taipei 110, Taiwan; 5School of Pharmacy, Taipei Medical University, Taipei 110, Taiwan; 6Graduate Institute of Biomedical Materials and Tissue Engineering, Taipei Medical University, Taipei 110, Taiwan; tonyliu@tmu.edu.tw; 7Department of Microbiology, Immunology and Biopharmaceuticals, College of Life Sciences, National Chiayi University, Chiayi 600, Taiwan

**Keywords:** acetylcholinesterase, passion fruits (*Passiflora edulis*), passive avoidance test, piceatannol, scirpusin B, scopolamine, seeds and rinds, sustainable utility

## Abstract

In this paper, the seeds and rinds of passion fruit, which are the agricultural waste of juice processing, were recycled to investigate their biological activities for sustainable use. De-oiled seed powders (S) were successively extracted by refluxing 95% ethanol (95E), 50E, and hot water (HW), respectively, to obtain S-95EE, S-50EE, and S-HWE. Dried rind powders were successively extracted by refluxing HW and 95E to obtain rind-HWE and rind-95EE, respectively. S-50EE and S-95EE showed the most potent extracts, such as anti-amyloid-β_1-42_ aggregations and anti-acetylcholinesterase inhibitors, and they exhibited neuroprotective activities against amyloid-β_25-35_-treated or H_2_O_2_-treated SH-SY5Y cells. Scirpusin B and piceatannol were identified in S-95EE, S-50EE, and rind-HWE, and they showed anti-acetylcholinesterase activity at 50% inhibitory concentrations of 62.9 and 258.9 μM, respectively. Daily pretreatments of de-oiled seed powders and rind-HWE (600 mg/kg), S-95EE, and S-50EE (250 mg/kg) or scirpusin B (40 mg/kg) for 7 days resulted in improved learning behavior in passive avoidance tests and had significant differences (*p* < 0.05) compared with those of the control in scopolamine-induced ICR mice. The seeds and rinds of passion fruit will be recycled as materials for the development of functional foods, promoting neuroprotection and delaying the onset of cognitive dysfunctions.

## 1. Introduction

The prevalence of dementia is increasing by nearly 10 million new cases every year, and more than 55 million people are believed to live with dementia globally in 2023, among which, Alzheimer’s disease (AD) is given more attention because it accounts for 60% to 70% of dementia cases in World Health Organization (WHO) reports [1]. The WHO has released statistical data showing that deaths caused by “AD and other dementias” dramatically jumped and ranked in the 7th position in 2019 [2], whereas the same item was 20th in 2000 and 14th in 2010 [3]. It has been statistically proven that older age is the greatest risk factor in AD; the others are genetics (APOE ε4 gene) and family history [4,5]. Recent studies have proposed that brain changes (or neuron damage) from AD begin 20 years or more before memory loss, and the number of damaged neurons and affected areas of the brain determine the symptoms of AD progression [4,5,6,7].

The pathologies of AD are characterized by synaptic failures with low levels of acetylcholine and neuron death, amyloid-like fibrils via amyloid-β (Aβ) peptide aggregations, and neurofibrillary tangles via hyper-phosphorylated Tau protein aggregations [8,9,10]. At present, at least three mechanism-based drugs have been approved by the US FDA to lessen the symptoms of AD patients or delay the progression of cognitive impairment in early-onset AD patients [11]. One class is acetylcholinesterase (AChE) inhibitors, based on the cholinergic hypothesis [9], such as donepezil [11,12]; another includes *N*-methyl-D-aspartate receptor antagonists (or glutamate regulators) such as memantine [11,13]; and the third is amyloid-targeting monoclonal antibodies such as lecanemab, approved in 2023 [11,14]. The relevant 18-month phase-III trial was an anti-Aβ-soluble protofibril antibody therapy administering intravenous lecanemab (10 mg/kg of body weight) or a placebo every two weeks; the lecanemab treatment was shown to result in 25% less cognitive decline compared with the placebo [14], and this was the first drug tested in the past 30 years to effectively delay the progression of cognitive impairment in early AD [10].

Reactive oxygen species (ROS) are closely associated with chronic diseases, such as cardiovascular and neurodegenerative diseases. The “free radical theory of aging” was originally proposed by Harman in 1956, in which the accumulation of free radicals during aerobic respiration results in cell aging and then death [15,16]. The endogenous and environmentally induced ROS in brain cells play accelerated roles in mitochondrial dysfunction, Aβ peptide aggregation via non-enzymatic glycation, lipid peroxidation, and microglia activation [17], which is linked to impaired cognition in AD and ROS-mediated aging in the brain. The ROS-scavenging activities of polyphenol-rich fruits, vegetables, and herbal medicines have been reported to perform anti-aging and neuroprotective functions [18,19]. Dietary polyphenols have been suggested for both in vitro biological activity assays and in vivo animal experiments and/or clinical trials, which may be conducted simultaneously to investigate the therapeutic potential of plants or their components in chronic diseases [18,19,20,21].

Extracts from the agricultural waste of water caltrop (*Trapa taiwanesis* Nakai) hulls and the active constituent of tellimagrandin II have shown in vitro antioxidant activities and dose-dependent anti-AChE activities, along with inhibitions against Aβ peptide (1-42) aggregations in vitro. Cognitive improvement in scopolamine-induced ICR mice with dysfunction has been evaluated with water maze and passive avoidance tests in vivo [22]. Piceatannol (a resveratrol analog, also called 3′-hydroxyresveratol) was reported to exhibit in vitro DPPH-radical-scavenging activity, anti-AChE activities, inhibitions against Aβ peptide (1-42) aggregations, and neuroprotective activities against Aβ peptide (25-35)-induced cell death in SH-SY5Y cell models, resulting in a reduction in cognitive impairment in scopolamine-induced amnesiac ICR mice [23]. The extracts of herbal plants (*Vitis thunbergii* var. *taiwaniana*) and the active constituent in vitisin A, a resveratrol tetramer, showed anti-AChE activities and neuroprotective activities against methylglyoxal-induced cell death in SH-SY5Y cell models, with resulting improvement in the scopolamine-induced cognitive dysfunction of amnesiac ICR mice evaluated using passive avoidance tests in vivo [24].

*Passiflora edulis* Sims, also called passion fruit, purple granadilla, or egg fruit, belongs to the Passifloraceae family, is native to tropical America, and has more than 500 species of which at least 50 or more are edible [25]. The main passion fruit cultivar in Taiwan is *Passiflora edulis* var. Tainung No. 1, a hybrid of *Passiflora edulis* (purple passion fruit) and *P. edulis* f. *flavicarpa* (yellow passion fruit). Nantou County, located in central Taiwan, is the largest cultivation area, accounting for about 75% of the total production in Taiwan [26,27]. Passion fruit harvested during summer is an economical fruit with high nutritional value [28]. It has been reported that quercetin and vanillic acid are the most abundant polyphenolic compounds in the pulp of passion fruit [29]. Passion fruit pulp contains high amounts of carotenoids and flavonoids [30]. Vast numbers of the seeds and rinds (or peels) of passion fruit are juice-processing by-products that are landfilled in the local downtown or prepared as compost, and if they are not properly reused, this agricultural waste can cause environmental pollution. The fresh peels and peel flours of passion fruit contain high amounts of dietary fiber [28,31]. The ethanol extracts of the dried rind powders of passion fruit contain dietary fiber, pectin, and flavonoids [32]. Flavonoids and anthocyanin (cyanidin-3-*O*-glucoside, peonidin-3-*O*-glucoside, and pelargonidin-3-*O*-glucoside) are found in the peel of purple and yellow passion fruit [33]. The intake of passion fruit rind flours by dextran sodium sulfate-induced C57BL/6 mice has caused anti-inflammatory effects, thereby maintaining the intestinal barrier and reducing colon damage, which are possibly the beneficial effects of the enriched dietary fibers and polyphenols in passion fruit rind flours [34]. The peels of passion fruit are also reported to exhibit anti-hypertension activities in spontaneously hypertensive rat models [35] and anti-obesity activities in reducing body weight gain and the weight ratio (g/100 g body weight) of retroperitoneal adipose tissue and epididymal adipose tissue in high-fat-diet-induced rat models [36]. The seeds of passion fruit contain high amounts of linoleic acid, oleic acid, palmitic acid, and protein [28,31]. The identified flavonoids in the ethanol extracts of passion fruit seeds include genistein, quercetin, and protocatechuic acid [31,37]. The two main stilbenoids of piceatannol and scirpusin B (piceatannol dimer), isolated by defatting the ethanol extracts of passion fruit seeds, have shown nitric oxide-dependent vasorelaxation activities in rat aorta blood vessel models [38]. Piceatannol and resveratrol isolated from fractions of the ethanol extracts of passion fruit seeds have been shown to reduce melanin synthesis and elevate collagen synthesis in dermal cell models [39]. Isolated scirpusin B and piceatannol from the seeds of passion fruit have shown potential α-glucosidase activities [40]. Previously, a commercial source of piceatannol was reported to impact AD-associated factors in vitro and reduce cognitive impairments in scopolamine-induced amnesiac ICR mice [23]. To take advantage of the additional value of the by-products of these seeds and rinds for sustainable uses, in this study, they were recovered from a juice-processing factory as raw materials and washed and dried for further extractions to investigate the effects of extracts or purified compounds on AD-associated factors in vitro and in vivo. The present results reveal that the seeds and rinds of passion fruit can be recycled for sustainable use to develop functional foods that promote neuroprotection and delay the onset of cognitive dysfunction.

## 2. Materials and Methods

### 2.1. Chemical and Reagents

The Thermo Scientific™ Pierce™ BCA Protein Assay Kit (Catalog No. PI23225) was purchased from ThermoFisher Scientific Inc. (Rockford, IL, USA). The protein rhAChE (recombinant human acetylcholinesterase/AChE, catalog number: 7574-CE) was purchased from R&D Systems Inc. (Minneapolis, MN, USA). The Aβ_1-42_ peptide and the Aβ_25-35_ peptide, thioflavin T (ThT, T3516), acetylthiocholine iodide (A5751), 5,5′-dithiobis(2-nitrobenzoic acid) (DTNB, D8130), 3-(4,5-dimethyl-2-thiazolyl)-2,5-diphenyl-2H-tetrazolium bromide (MTT, 475989), 2,2-diphenyl-1-picrylhydrazyl (DPPH, D9132), bovine serum albumin (BSA), poly-L-Lysine solution, and scopolamine hydrobromide (S0929) were purchased from Sigma Chemical Co. (St. Louis, MO, USA). Fetal bovine serum (FBS), Dulbecco’s modified Eagle medium (DMEM), and DMEM/F-12 medium were purchased from Gibco BRL Life Technologies (Grand Island, NY, USA).

### 2.2. Extraction of Seeds and Rinds of Passion Fruits

The seeds and rinds of passion fruits (*Passiflora edulis* var. Tainung No. 1) were received in February 2022 from a local passion-fruit-processing factory in Puli Town (Nantou country, Taiwan) via a frozen delivery. After being washed with tap water and dried in an oven at 45 °C for three days, the seeds and rinds were separately extracted as follows (Figure 1). The dried, powdered seeds of passion fruits (1.2 kg) were put into a 5 L round bottle flask and extracted with refluxing *n*-hexane (3.5 L) in a cooling system for two hours. After being filtrated, the residues were repeatedly extracted with *n*-hexane. The *n*-hexane was removed with reduced pressure in a rotary evaporator to obtain hexane-extractable seed oil (recovery of 18.61–21%), and the residues were placed in a hood overnight to obtain de-oiled seed powders (S, recovery of 77.5–79.25%). The S was then extracted via refluxing with 12 L of 95% ethanol (95E) at 90 °C for two hours twice to obtain 95E extracts (S-95EE); the residue was extracted via refluxing with 12 L of 50% ethanol (50E) at 90 °C for two hours twice to obtain S-50EE; the residue was extracted via refluxing with 12 L of boiling water (HW) for two hours twice to obtain S-HWE. Each filtrate was concentrated with reduced pressure in a rotary evaporator and then lyophilized to obtain S-95EE (86.9 g), S-50EE (42.9 g), and S-HWE (21.6 g), respectively. These seed extracts were kept in a refrigerator before use. The dried, pulverized rinds of passion fruits (600 g) were extracted 20 times (*w*/*v*) with boiling water for two hours twice. The filtrate was concentrated with reduced pressure in a rotary evaporator and lyophilized to obtain hot water extract (rind-HWE). The residue was dried in an oven at 60 °C and extracted 10 times (*w*/*v*) by refluxing 95E at 90 °C for two hours twice to obtain 95E extracts (rind-95EE). Each filtrate was concentrated with reduced pressure in a rotary evaporator and then lyophilized to obtain rind-HWE (292.05 g) and rind-95EE (10.25 g). These rind extracts were kept in a refrigerator before use.

### 2.3. Isolation and Identification of Piceatannol and Scirpusin B from Seed Extracts of Passion Fruits and HPLC Fingerprinting Analyses

S-50EE (30.1 g) was dissolved in methanol and loaded on a Sephadex LH-20 column (5.0 cm i.d. × 49 cm). The absorbed samples were eluted with methanol to obtain four fractions, and each was concentrated with reduced pressure in a rotary evaporator and then lyophilized to obtain S-50EE-1 (9.7 g), S-50EE-2 (4.9 g), S-50EE-3 (4.4 g), and S-50EE-4 (2.7 g). The S-50EE-2 fraction was performed on a LiChroprep RP-18 (2.5 cm i.d. × 45.5 cm) column and eluted with a mixture of 0.05% trifluoroacetic acid (TFA) and acetonitrile (CH_3_CN) at a fixed ratio of 75:25 to obtain piceatannol (25 mg, purity 98.1%). The S-510EE-3 fraction was performed on a LiChroprep RP-18 (2.5 cm i.d. × 45.5 cm) column and eluted with a mixture of 0.05% TFA and CH_3_CN at a fixed ratio of 75:25. Fractions 25–45 were collected and concentrated, then loaded on a Sephadex LH-20 column (2.5 cm i.d. × 45.5 cm), and then eluted with methanol. Fractions 42–49 were collected and concentrated, then loaded again on a LiChroprep RP-18 (2.5 cm i.d. × 45.5 cm) column, and then eluted with a mixture of 0.05% TFA and CH_3_CN at a fixed ratio of 75:25 to obtain scirpusin B (216 mg, purity 99.6%). Structural elucidations of the isolated compounds were determined using ^1^H- and ^13^C-NMR and compared with data from the literature [38,41].

Piceatannol (**1**): ^1^H-NMR (500 MHz, CD_3_OD), δ: 6.97 (1H, d, *J* = 1.9 Hz, H-2), 6.89 (1H, d, *J* = 16.2 Hz, H-7), 6.83 (1H, dd, *J* = 1.9, 8.2 Hz, H-6), 6.74 (1H, d, *J* = 16.2 Hz, H-5), 6.73 (1H, d, *J* = 8.2 Hz, H-8), 6.43 (2H, d, *J* = 2.1 Hz, H-2′, H-6′), 6.15 (1H, t, *J* = 2.1 Hz, H-4′). ^13^C-NMR (150 MHz, CD_3_OD), δ: 159.8 (C-3′, C-5′), 146.68 (C-4), 146.65 (C-3), 141.4 (C-1′), 131.2 (C-1), 129.8 (C-7), 127.2 (C-8), 120.3 (C-6), 116.6 (C-4), 114.0 (C-2), 105.9 (C-2′, C-6′), 102.8 (C-4′).

Scirpusin B (**2**): ^1^H-NMR (500 MHz, CD_3_OD), δ: 6.77 (1H, d, *J* = 16.0 Hz, H-7), 6.76 (1H, d, *J* = 2.0 Hz, H-2′), 6.74 (1H, d, *J* = 8.0 Hz, H-5′), 6.70 (1H, d, *J* = 2.0 Hz, H-2), 6.65 (2H, *J* = 8.4 Hz, H-5, H-6′), 6.63 (1H, brs, H-14), 6.58 (1H, dd, *J* = 8.4, 2.0 Hz, H-6), 6.54 (1H, d, *J* = 16.0 Hz, H-8), 6.26 (1H, brs, H-12), 6.18 (1H, t, *J* = 2.0 Hz, H-12′), 6.15 (2H, d, *J* = 2.0 Hz, H-10′), 5.28 (1H, d, *J* = 5.8 Hz, H-7′), 4.34 (1H, d, *J* = 5.8 Hz, H-8′). ^13^C-NMR (125 MHz, CD_3_OD) δ: 162.97 (C-11), 160.08, 160.03 (C-11′, C-13′), 159.83 (C-13), 147.78 (C-9′), 146.72, 146.62 (C-4, C-3), 146.54, 146.43 (C-4′, C-3′), 137.10 (C-9), 135.09 (C-1′), 131.13, 131.04 (C-1, C-7), 123.76 (C-8), 120.17 (C-6), 119.96 (C-10), 118.60 (C-6′), 116.49, 116.42 (C-5, C-5′), 114.22 (C-2), 113.81 (C-2′), 107.48 (C-10′, C-14′), 104.43 (C-14), 102.39 (C-12′), 96.96 (C-12), 95.02 (C-7′), 58.19 (C-8′).

For fingerprinting and quantitative analyses, piceatannol and scirpusin B (1.0 mg) were individually dissolved in 1 mL of MeOH as the standard solution, and each extract was dissolved in MeOH to 10 mg/mL. The analytical LiChrospher 100 RP-18e (4 mm i.d. × 250 mm, 5 µm) HPLC column used the Waters Chromatography system (Waters Co., Milford, MA, USA). A gradient elution program for the mobile phase was set as follows: 0.05% TFA:CH_3_CN (0 min, 95:5; 45 min, 0:100; 55 min, 0:100; 56 min, 95:5; 65 min, 95:5). The mobile phase was pumped at a flow rate of 1.0 mL/min, and 10 μL was injected for analysis. The column temperature was kept at 40 °C, and the wavelength was set at 320 nm for monitoring. The piceatannol and scirpusin B contents in each extract were calculated based on the peak area compared with standard solutions in the same HPLC conditions, as per the following equation:
Content (%) = 10 (A_E_/A_S_)

A_E_: The peak area of scirpusin B (or piceatannol) in extracts.

A_S_: The peak area of scirpusin B (or piceatannol) in the standard solution.

### 2.4. Biological Activity Assays In Vitro

#### 2.4.1. DPPH Radical-Scavenging Activities

The DPPH radical-scavenging activities of the seed and rind extracts of passion fruit were determined according to previous reports [22,23,42] with some modifications. Each 50 μL of the sample solution (S-95EE and S-50EE, 1.25, 2.5, 5, and 10 μg/mL; S-HWE, 100, 200, and 400 μg/mL; rind-HWE and rind-95EE, 100, 125, 250, and 500 μg/mL) in dimethyl sulfoxide (DMSO) was placed into a 96-well plate, and then 150 μL of 120 μM DPPH in methanol was added. The mixture was reacted at 37 °C for 20 min under light protection. The decrease in absorbance at 515 nm was measured and expressed as ΔA515 nm. The aliquot DMSO was used instead of the sample solution in the blank. The DPPH radical scavenging activity (%) was calculated with the following equation: (ΔA515_blank_ − ΔA515_sample_) ÷ △A515_blank_ × 100%. The 50% inhibitory concentration (IC_50_) of the DPPH radical scavenging activity was calculated from each linear equation: for S-95EE, 2.5, 5, and 10 μg/mL were used; for S-50EE, 1.25, 2.5, and 5 μg/mL were used; for S-HWE, 100, 200, and 400 μg/mL were used; for rind-HWE and rind-95EE, 100, 125, and 250 μg/mL were used.

#### 2.4.2. AChE-Inhibitory Activities In Vitro

The AChE-inhibitory activities of different extracts of the seeds and rinds of passion fruit were determined following previous reports [22,23,24,43]. Each extract was dissolved in DMSO (S-95EE, 0.04, 0.1, 0.2, 0.4 mg/mL; S-50EE, 0.01, 0.025, 0.05, and 0.1 mg/mL; S-HWE, 0.4, 1, 2, and 4 mg/mL; rind-HWE, 4, 10, 20, and 40 mg/mL; rind-95EE, 4, 8, 15, and 32 mg/mL) for comparison. The purified scirpusin B (12.5, 25, 50, and 100 μM) and piceatannol (50, 100, 200, 400, and 800 μM) were dissolved in DMSO and used for comparison. An equal aliquot of DMSO was used in the blank instead of the sample solution. The free thiol group of the released thiocholine from AChE-hydrolyzed acetylthiocholine was coupled with DTNB to generate yellowish products, and the absorbance at 405 nm was recorded for 10 min. AChE inhibition (%) was calculated as follows: [(A405_blank_ − A405_sample_)/(A405_blank_)] × 100%. The IC_50_ of AChE was calculated from each linear equation: for S-95EE, 0.04, 0.1, and 0.2 mg/mL were used; for S-50EE, 0.01, 0.025, and 0.05 mg/mL were used; for S-HWE, 1, 2, and 4 mg/mL were used; for rind-HWE, 4, 10, and 20 mg/mL were used; for rind-95EE, 4, 8, and 16 mg/mL were used; for scirpusin B, 25, 50, and 100 μM were used; and for piceatannol, 200, 400, and 800 μM were used.

#### 2.4.3. Inhibition against Aβ Peptide Aggregations In Vitro

The anti-Aβ_1-42_ peptide aggregations of different extracts of passion fruit seeds and rinds were determined following previous reports [22,23,43] by monitoring fluorescent changes in ThT-binding self-aggregated Aβ fibrils [44]. The preparation of the Aβ_1-42_ peptide stock solution for self-aggregations was conducted according to a previous report [23]. The Aβ_1-42_ peptide working solution (10 μM) was mixed with 10 μM of ThT solution and each extract (S-95EE and S-50EE, 0.5, 1, 2, and 5 μg/mL; S-HWE, 1, 2, 5, and 10 μg/mL; rind-HWE, 25, 50, and 100 μg/mL; rind-95EE, 2, 5, 10, and 20 μg/mL). The purified scirpusin B and piceatannol (0.05, 0.1, 0.2, 0.5, and 1.0 μM) were dissolved in DMSO and used for comparison. An equal aliquot of DMSO was used in the control instead of the sample solution. The mixture was shaken continuously at 37 °C for 24 h. The excitation was set at 440 nm, emission was set at 486 nm, and the Ex/Em ratio (E) was determined at the beginning (0 h) and the end (24 h) of the reaction. The inhibition against Aβ aggregation (%) was calculated as [(E_control,24-h_ − E_control,0-h_) − (E_sample,24-h_ − E_sample,0-h_)/(E_control,24h_ − E_control,0-h_)] × 100%. The IC_50_ of anti-Aβ_1-42_ peptide aggregation was calculated from each linear equation: for S-95EE and S-50EE, 0.5, 1, and 2 μg/mL were used; for S-HWE, 2, 5, and 10 μg/mL were used; for rind-HWE, 25, 50, and 100 μg/mL were used; for rind-95EE, 2, 5, and 10 μg/mL were used; and for scirpusin B and piceatannol, 0.2, 0.5, and 1.0 μM were used.

#### 2.4.4. Neuroprotection against Hydrogen Peroxide-Induced or Aβ_25-35_ Peptide-Induced Cell Death in SH-SY5Y Cell Models

The SH-SY5Y cells were used as in vitro models (as human neuroblastoma cell lines from the American Type Culture Collection, Manassas, VA, USA) to investigate the neuroprotective activities of different extracts of the seeds and rinds of passion fruits against hydrogen peroxide (100 μM)-induced or Aβ_25-35_ peptide (40 μg/mL)-induced cell death following a previous report [23]. The cells (1 × 10^4^ cells/well) were seeded onto a 96-well microplate and cultured in a DMEM/F-12 medium containing 10% FBS for 24 h at 37 °C under a humidified atmosphere and 5% CO_2_. The culture medium was removed from the cultured plate, and different concentrations (12.5, 25, and 50 μg/mL) of extracts of seeds and rinds of passion fruits or 0.1% DMSO (the control and the blank) were added and incubated at 37 °C in a humidified atmosphere with 5% CO_2_ for 24 h. Different concentrations of purified scirpusin B and piceatannol (0.5, 1.0, 2.0, and 5.0 μM) were added and incubated at 37 °C in a humidified atmosphere with 5% CO_2_ for 24 h. The treated medium was removed, washed with PBS, and 40 μg/mL of Aβ_25-35_ peptide in PBS or 100 μM hydrogen peroxide in PBS was added to the medium for another 24 h culture. An equal aliquot of PBS was added to the blank. Then, MTT was used to evaluate cell viability, and the absorbance at 570 nm was determined [45,46]. The blank was a 0.1% DMSO addition; the control was a (0.1% DMSO + 100 μM hydrogen peroxide) or (0.1% DMSO + 40 μg/mL amyloid-β_25-35_ peptide) addition. The blank was recognized as 100%, and each treatment was expressed as the relative cell viability (%). Before MTT assays, the cell morphologies in hydrogen peroxide-treated SH-SY5Y cells with or without pretreatments of extracts of seeds and rinds of passion fruits were photographed using an inverted microscope (400-fold magnification, ECLIPSE TS100, Nikon Instruments Inc., Tokyo, Japan).

### 2.5. Molecular Docking in Silico

Molecular docking analyses were performed using the CDOCKER program in Discovery Studio (DS), a BIOVIA program, to analyze AChE and Aβ_1-42_ peptide with scirpusin B following a previous report [23]. Information on the crystal structures of AChE (https://www.rcsb.org/structure/4ey7 (accessed on 1 September 2023)) and the Aβ_1-42_ peptide (https://www.rcsb.org/structure/1z0q (accessed on 1 September 2023)) from the protein data bank were selected in docking studies [47] using the automatic ligand preparation function in the Macromolecules Tools of DS. The AChE binding site was defined as 10 Å from the co-crystalized ligand. The Aβ_1-42_ peptide binding site was detected automatically with a 10 Å radius sphere. A docking compound of scirpusin B or piceatannol was prepared with the ligand preparation function in the Small Molecules Tools of DS.

### 2.6. Effects of Extracts of Seeds and Rinds of Passion Fruits or Scirpuin B Pretreatments on Cognitive Dysfunctions in Scopolamine-Induced Amnesiac ICR Mice

Animal experiments for learning dysfunction models were conducted following previous reports [22,23,24,43] by using scopolamine-induced ICR mice based on the cholinergic hypothesis [9]. The protocols of the animal experiments are Figure 1 and Figure 2.

#### 2.6.1. Effects of 7-Day Pretreatments with De-Oiled Seed Powders (S), S-95EE, S-50EE, or Rind-HWE on the Improvement of Learning and Memory Functions

Six-week-old male ICR mice were purchased from the Laboratory Animal Center (National Taiwan University, Taipei). The approval number of the animal experimental protocols was LAC-2021-0488. The protocol is shown in Figure 1. After one week of acclimation, the mice were randomly divided into 7 groups (N = 5 for each group), including blank, control, positive control (donepezil administration), and four pretreatment (de-oiled seed powders (S) and rind-HWE, 600 mg/kg; S-95EE and S-50EE, 250 mg/kg) groups. In the first pre-oral stage, mice in the four pretreatment groups were administered each sample with an oral gavage once a day for 7 days; mice in the blank, control, and positive control groups were orally administered the same volume of distilled water for 7 days. On day 8 and day 9, the second oral stage (treatment and learning behavior evaluations), mice in the pretreatment group (S or rind-HWE, 600 mg/kg; S-95EE or S-50EE, 250 mg/kg) and positive control group (donepezil, 5 mg/kg) were each administered daily with an oral gavage; each mouse in the blank and the control was treated orally with an equal volume of distilled water. After 30 min of oral administration, scopolamine (1 mg/kg dissolved in PBS) was delivered to each mouse with an intraperitoneal injection. Thirty minutes after the scopolamine injection, the learning behavior of each mouse was evaluated with a passive avoidance test. Each mouse in the blank was injected with PBS and then evaluated for learning behavior.

#### 2.6.2. Effects of 18-Day Pretreatments with De-Oiled Seed Powders (S) or Rind-HWE on the Improvement of Learning and Memory Functions

Six-week-old male ICR mice were purchased from the Laboratory Animal Center (National Taiwan University, Taipei). The approval number of the animal experimental protocols was LAC-2021-0488. The protocol is shown in Figure 2. After one week of acclimation, the mice were randomly divided into 7 groups (N = 5 for each group), including blank, control, positive control (donepezil administration), and four pretreatment (de-oiled seed powders (S), 250 and 400 mg/kg and rind-HWE, 250 and 400 mg/kg) groups. In the first pre-oral stage, mice in the four pretreatment groups were administered each sample with an oral gavage once a day for 18 days; mice in the blank, control, and positive control groups were orally administered the same volume of distilled water for 18 days. On day 19 and day 20 of the second oral stage (treatment and learning behavior evaluations), mice in each group were treated the same as those in the Section 2.6.1.

#### 2.6.3. Effects of 7-Day Pretreatments with Scirpusin B or Piceatannol on the Improvement in Learning and Memory Functions

Six-week-old male ICR mice were purchased from the Laboratory Animal Center (National Taiwan University, Taipei). The approval number of the animal experimental protocols was LAC-2021-0488. The protocol is shown in Figure 1. After one week of acclimation, the mice were randomly divided into 5 groups (N = 5 for each group), including blank, control, positive control (donepezil administration), and two pretreatment (scirpusin B (40 mg/kg) or piceatannol (50 mg/kg)) groups. In the first pre-oral stage, mice in the two pretreatment groups were administered each sample with an oral gavage once a day for 7 days; mice in the blank, control, and positive control groups were orally administered the same volume of distilled water for 7 days. On day 8 and day 9 of the second oral stage (treatment and learning behavior evaluation), mice in each group were treated the same as those in the Section 2.6.1. All the mice were sacrificed at the end of the animal experiments, and the brain tissue of the mice in each group was isolated, weighed, and stored at −80 °C for AChE activity assays. All the mouse brain tissue was weighed and powdered in liquid nitrogen using a mortar and pestle, suspended, and extracted with 0.5 to 1 mL of tenfold-diluted PBS. The AChE activities (normalized with protein contents) in the extracts of each group were assayed with the modifying methods described in Section 2.4.2. The protein content in all the mouse brain extracts was quantified with a BCA protein kit using BSA to plot a standard curve. For AChE activity determinations in the mice brain extracts, the whole mouse brain extracts were pre-mixed with 100 nM of donepezil solution (in the final concentration) for 30 min as a sample blank (the DTNB-reacting substances but not AChE-generated DTNB-reacting thiocholine) in parallel with AChE activity assays in each brain extract sample, as described in Section 2.4.2. The AChE activity was calculated as [A405_sample_ − A405_sample blank_]/μg protein_sample_, and AChE activity in the blank group was identified as 100%.

### 2.7. The Learning Dysfunction of Scopolamine-Induced ICR Mice in Passive Avoidance Tests

The passive avoidance test, including an acquisition trial and a retention trial, was used to investigate the effects of the oral administrations of the seed and rind extracts of passion fruit and purified compounds on learning dysfunction in scopolamine-induced ICR mice following previous reports [22,23,24,43]. The passive/active avoidance computerized apparatus (PACS-30, Columbus Instruments Inc., Columbus, OH, USA) contained two connected chambers with an LED lightbox and a dark box separated by a sliding guillotine door. On the first day of the acquisition trial, each mouse was placed in the LED lightbox. The mouse entered the dark box, and at the same time, the sliding guillotine door was closed and the time in the lightbox (the step-through latency, s) was recorded. Each mouse in the dark box received an electric foot shock (0.3 mA for 3 s) delivered with a wired metal floor and then sent back to its cage. On the second day of the retention trial (when the electric foot shock in the dark box was shut down), each mouse was placed again in the LED lightbox, and the time in the lightbox (the step-through latency, sec) was recorded.

### 2.8. Statistical Analyses

The present data are expressed as the mean ± SD of three independent experiments, and the step-through latency in the passive avoidance test of each animal experiment is expressed as mean ± SE. Student’s *t*-test was used to compare neuroprotective activities against hydrogen peroxide-induced or Aβ_25-35_-induced SH-Y5Y cell death between [(the control) vs. (each sample treatment)] or [(the control) vs. (the blank)]; Student’s *t*-test was also used to compare the step-through latency (s) of pretreatments of crude extracts (Section 2.6.1 and Section 2.6.2) between [(the scopolamine-induced control) vs. (each sample pretreatment)] or [(the scopolamine-induced control) vs. (the blank)]. It was considered a statistically significant difference when *p* < 0.05 *, *p* < 0.01 **, or *p* < 0.001 ***. One-way analysis of variance (ANOVA) and a post hoc Tukey’s test were used to analyze multiple group comparisons of step-through latency (s) and brain extracts for AChE activities among pretreatments of isolated pure compounds (Section 2.6.3) in scopolamine-induced ICR mice. It was considered a statistically significant difference (*p* < 0.05) when there were different uppercase letters (for the retention trial or the AChE activity in the brain extracts) or lowercase letters (for the acquisition trial) in each bar. The GraphPad Prism Window 6.0 software (San Diego, CA, USA) was used to perform statistical analyses.

## 3. Results

### 3.1. Biological Activities of Extracts of Seeds and Rinds in Vitro

Figure 2A shows the DPPH radical-scavenging activities. It was found that the extracts of seeds and rinds showed DPPH radical-scavenging activities to varying degrees. The IC_50_ values of the DPPH radical-scavenging activities of S-95EE, S-50EE, and S-HWE were 7.01, 3.28, and 143 μg/mL, respectively. The IC_50_ values of the DPPH radical-scavenging activities of rind-HWE and rind-95EE were 122.08 and 105.94 μg/mL, respectively. Figure 2B shows the AChE-inhibitory activities in vitro. It was found that seed and rind extracts showed anti-AChE activities to varying degrees. The IC_50_ values of the anti-AChE activities of S-95EE, S-50EE, and S-HWE were 74.3, 25.7, and higher than 100 μg/mL, respectively. The IC_50_ of the anti-AChE activities of rind-HWE and rind-95EE was higher than 100 μg/mL. Figure 2C shows the anti-Aβ_1-42_ peptide aggregations in vitro. The IC_50_ values of the anti-Aβ_1-42_ peptide aggregations of S-95EE, S-50EE, and S-HWE were 0.82, 0.45, and 5.15 μg/mL, respectively. The IC_50_ values of the anti-Aβ_1-42_ peptide aggregations of rind-HWE and rind-95EE were 98.73 and 6.84 μg/mL, respectively. Figure 2D shows neuroprotective activities against H_2_O_2_ (100 μM)-induced cell death in SH-SY5Y cell models. The treatment with 100 μM of H_2_O_2_ showed a reduction and had a significant difference (*p* < 0.001) in cell viability from 100% (the blank) to about 80% (the control). The pretreatment with different concentrations of S-95EE, S-50EE, S-HWE, and rind-HWE, but not rind-95EE, elevated the cell viabilities of H_2_O_2_-treated SH-SY5Y cells and had significant differences compared with the control (*p* < 0.01 **, 0.001 ***). It was noted that the H_2_O_2_-treated cells not only lowered cell viability but also shortened the neurite lengths of SH-SY5Y cells under an inverted microscope (400-fold magnification, Appendix A). The pretreatment with 50 μg/mL of S-95EE, S-50EE, S-HWE, rind-HWE, or rind-95EE was shown to recover the shortened neurite lengths in H_2_O_2_-treated SH-SY5Y cells (Appendix A). Figure 2E shows neuroprotective activities against Aβ_25-35_ peptide (40 μg/mL)-induced cell death in SH-SY5Y cell models. The treatment with 40 μg/mL Aβ_25-35_ peptide showed a reduction in cell viability and had a significant difference (*p* < 0.001) from 100% (the blank) to 68.2% (the control). The pretreatments with S-95EE, S-50EE, and S-HWE at concentrations of 12.5, 25, and 50 μg/mL were shown to elevate the viability of SH-SY5Y cells by reducing the toxicity of the Aβ_25-35_ peptide and showed significant differences from the treated Aβ_25-35_ peptide only (the control, *p* < 0.05 *, 0.01 **, 0.001 ***). The pretreatment with rind-HWE at concentrations of 25 and 50 μg/mL elevated the viability of SH-SY5Y cells by reducing the toxicity of the Aβ_25-35_ peptide and showed significant differences from the treated Aβ_25-35_ peptide only (the control, *p* < 0.01 **, 0.001 ***).

### 3.2. Biological Activities of Isolated Piceatannol and Scirpusin B

Figure 3A shows the structures of isolated piceatannol and scirpusin B from S-50EE. Scirpusin B is the dimer of piceatannol. Figure 3B shows the AChE-inhibitory activities of isolated piceatannol and scirpusin B in vitro. It was found that the isolated piceatannol and scirpusin B showed potential anti-AChE activities. The IC_50_ values of the anti-AChE activities of piceatannol and scirpusin B were 258.9 and 62.9 μM, respectively. Scirpusin B showed fourfold greater AChE-inhibitory activity than piceatannol. Figure 3C shows the anti-Aβ_1-42_ peptide aggregations of isolated piceatannol and scirpusin B in vitro. It was found that the isolated piceatannol and scirpusin B showed potential anti-Aβ_1-42_ peptide aggregations. The IC_50_ values of the anti-Aβ_1-42_ peptide aggregations of piceatannol and scirpusin B were 0.34 and 0.63 μM, respectively. Regarding the anti-Aβ_1-42_ peptide aggregations, the piceatannol showed twofold greater anti-Aβ_1-42_ peptide aggregations than scirpusin B. Figure 3D shows the neuroprotective activities of isolated piceatannol and scirpusin B against the Aβ_25-35_ peptide (40 μg/mL)-induced cell death in SH-SY5Y cell models. The treatment with 40 μg/mL of Aβ_25-35_ peptide showed a reduction in cell viability and had a significant difference (*p* < 0.001) from 100% (the blank) to 73.15% (the control). The pretreatments with scirpusin B and piceatannol at concentrations of 0.5, 1.0, 2.0, and 5.0 μM were shown to elevate the viability of SH-SY5Y cells by reducing the toxicity of the Aβ_25-35_ peptide and showed significant differences from the treated Aβ_25-35_ peptide only (the control, *p* < 0.01 **, 0.001 ***).

Figure 4 shows the HPLC profiles of (A) purified piceatannol (peak 1), (B) purified scirpusin B (peak 2), (C) S-95EE, (D) S-50EE, and (E) rind-HWE. The piceatannol (peak 1) and scirpusin B (peak 2) were both identified in the HPLC chromatograms of S-95EE, S-50EE, and rind-HWE by monitoring A320 nm. For fingerprinting and quantitative analyses, piceatannol and scirpusin B (1.0 mg) were each dissolved in 1 mL of MeOH as the standard solution, and each extract was dissolved in MeOH to 10 mg/mL. It was found that the three extracts of S-95EE, S-50EE, and rind-HWE all contained piceatannol (peak 1) and scirpusin B (peak 2) to varying degrees. The piceatannol and scirpusin B contents in S-95EE were 6.90% and 9.67%, respectively; in S-50EE, they were 2.51% and 16.99%, respectively; in rind-HWE, they were 0.34% and 0.51%, respectively. The extracts of the de-oiled seeds of passion fruit, S-95EE and S-50EE, contained higher amounts of scirpusin B than piceatannol. Both piceatannol and scirpusin B were the active components in S-95EE and S-50EE based on the abovementioned results for their AD-associated biological activities.

### 3.3. Molecular Dockings of Scirpusin B with AChE

Figure 5A shows the docking pose of scirpusin B (yellow) with AChE (light blue). Scirpusin B formed three hydrogen bonds (green dashed lines) with residues Asn87 (N87), Glu202 (E202), and His447 (H447). Scirpusin B showed hydrophobic interactions with key amino acids, such as π–π-stacked interactions with residues Trp86 (W86), W286, and Tyr341 (Y341); π–π T-shaped interactions with residues Y124 and Y337; and π–alkyl interactions with residues Y124 and Y341. Figure 5B shows the superimposed docking poses of donepezil (salmon) and scirpusin B (yellow) in the AChE active site. Scirpusin B and donepezil shared similar docking poses in the occupied AChE regions. Donepezil (as a reference compound) formed one hydrogen bond (green dashed lines) with residue Phe295 (F295). The donepezil/AChE and scirpusin B/AChE shared the same hydrophobic interactions with key amino acids, including a π–π-stacked interaction with residues W86, W286, and Y341 and π–alkyl interactions with residue Y341. Figure 5C shows the superimposed docking poses of piceatannol (cyan) and scirpusin B (yellow) in the AChE active site. Piceatannol could form three hydrogen bonds (green dashed lines) with residues F295, H447, and Gly122 (G122). Both scirpusin B/AChE and piceatannol/AChE shared the same hydrophobic interactions with key amino acids, including a π–π-stacked interaction with residue W286 and π–π T-shaped interactions with residue Y124. The docking poses of scirpusin B and piceatannol with AChE showed the same number of three hydrogen bonds but had different numbers of hydrophobic interactions. The interaction energy of the docking pose of scirpusin B/AChE was −39.6931 Kcal/mol, and the interaction energy of the docking pose of piceatannol/AChE was −36.0156 Kcal/mol. The more stable scirpusin B/AChE complex had a lower interaction energy compared with the piceatannol/AChE complex, which matched the higher AChE inhibitions of scirpusin B compared with piceatannol in a present study (Figure 3B).

### 3.4. Molecular Dockings of Scirpusin B with Aβ_1-42_ Monomer

Figure 6A shows the docking pose of scirpusin B (yellow) with Aβ monomer (gray). The higher molecular size of scirpusin B occupied the groove regions of residues 11 to 22 of the Aβ_1-42_ peptide helix. Scirpusin B could form one hydrogen bond (green dashed line) with residue Glu11 (E11) and two hydrogen bonds with residue E22, and it also formed π–alkyl and alkyl hydrophobic interactions with residue Val18 (V18). Figure 6B shows the superimposed docking poses of piceatannol (cyan) and scirpusin B (yellow) with Aβ monomer (gray). Piceatannol occupied the regions of residues 18 to 24 of the Aβ_1-42_ peptide helix and could form two hydrogen bonds with residue E22 and V24 and π–alkyl interactions with residue V18. The interaction energy of the docking pose of scirpusin B/Aβ_1-42_ at the regions of residues 11 to 22 was −35.6229 Kcal/mol, and the interaction energy of the docking pose of piceatannol/Aβ_1-42_ at the regions of residues 18 to 24 was −25.1388 Kcal/mol. The more stable complex of scirpusin B/Aβ_1-42_ at the regions of residues 11 to 22 had a lower interaction energy compared with the complex of piceatannol/Aβ_1-42_ at the regions of residues 18 to 24. It was reported that residues 16 to 21 of the Aβ_1-42_ monomer were the core segments of Aβ self-aggregations [48]. Based on the results of the anti-Aβ_1-42_ aggregations in Figure 3C, the docking poses of both piceatannol and scirpusin B occupied the core region of Aβ aggregations and were shown to be potent breakers retarding Aβ self-aggregations.

### 3.5. Animal Experiments in Scopolamine-Induced ICR Mice with Cognitive Dysfunctions

Figure 7A shows the results of 2-day learning behaviors, in which mice received high doses of different extracts, including S and rind-HWE (600 mg/kg) and S-95EE and S-50EE (250 mg/kg) in the 7-day pretreatment protocol (Figure 1). Mice treated with donepezil (5 mg/kg) 30 min before learning behavior evaluations were used as the positive control for comparisons. A staying time in the lightbox (the step-through latency, s) of two days was recorded. Mice in each group showed no significant difference (*p* > 0.05) compared with the control group in the step-through latency (s) on the first day of the acquisition trial (the black column and the red arrow); however, on the second day of the retention trial (gray column and the black arrow), mice in the pretreatment groups of S and rind-HWE (600 mg/kg), S-95EE and S-50EE (250 mg/kg), and the blank each showed a longer step-through latency (sec) compared with the control group and had significant differences (*p* < 0.05, 0.01, 0.001), as determined by Student’s *t*-test.

Owing to the high recovery of S from seeds (77.5–79.25%) and rind-HWE from rinds (around 48–50%), the dose effects of S and rind-HWE were measured in the 18-day pretreatment protocol (Figure 2). S and rind-HWE (250 and 400 mg/kg) were each administered to mice via gavage once a day for 18 days, and then, a 2-day passive avoidance test was administered to evaluate learning behavior (Figure 7B). There were no significant differences in the step-through latency (s) of the acquisition trial (the white column and the red arrow) between the treated groups and the control or the blank and the control (*p* > 0.05); however, on the second day of the retention trial (slash column and the black arrow), mice in the pretreatment groups of S (400 mg/kg) and rind-HWE (250 and 400 mg/kg) and mice in the blank, but not the S group, at 250 mg/kg each showed a longer step-through latency (s) and had significant differences (*p* < 0.05, 0.001) compared with the control group according to Student’s *t*-test. The results indicated that a lower dose for a longer pretreatment—S at 400 mg/kg for 18 days or rind-HWE at 250 and 400 mg/kg for 18 days—seemed to improve learning behaviors in mice with cognitive dysfunction induced by scopolamine.

Figure 7C shows the effects of piceatannol (50 mg/kg) and scirpusin B (40 mg/kg) on the learning behaviors of scopolamine-induced mice with a 7-day pretreatment protocol. It was found that the step-through latency (s) of mice in the acquisition trial showed no significant differences between the groups (*p* > 0.05), which were marked by the same lowercase alphabetic letter (analyzed with one-way ANOVA and a post hoc Tukey’s test). On the second day of the retention trial, the mice in the blank, the positive control of the donepezil-treated group, the piceatannol pre-treated group, and the scirpusin B pre-treated group showed a higher step-through latency in the lightbox and showed significant differences compared with the control (*p* < 0.05, analyzed with one-way ANOVA and a post hoc Tukey’s test), which was marked by a different uppercase alphabetic letter. This result indicated that scirpusin B and piceatannol were the active compounds of S-95EE and S-50EE, contributing to improved learning and memory functions. After the mice were sacrificed, the AChE activities (normalized with the extractable protein) in the brain extracts of different groups were determined and are shown in Figure 7D. As expected, AChE activities in the brain extracts of mice in the donepezil-treated group, the piceatannol pre-treated group, and the scirpusin B-pre-treated group showed significant reductions compared with those in the control (*p* < 0.05).

## 4. Discussion

This was the first report showing that the waste of passion fruit, seeds and rinds from a local juice-processing factory, could be used as ingredients to develop functional foods for daily intake providing neuroprotection and delaying the onset of cognitive dysfunctions. These in vitro data on the anti-AD-associated factors and in vivo learning behavior improvements from the de-oiled seed powders (S) rind-HWE, S-95EE, and S-50EE (Figure 7A,B) might be beneficial in sustainable use through the recycling of the seeds and rinds of passion fruit. At present, FDA-approved AChE inhibitors have shown short-term benefits for improving cognitive symptoms and have no benefits in delaying the progression of AD [9]; an Aβ protofibril-clearing antibody, lecanemab (10 mg/kg of body weight, every two weeks), was shown to reduce cognitive decline by 25% compared with a placebo in an 18-month phase-III trial for mild cognitive impairment or mild dementia caused by AD [14]. The poor outcomes of AD might be due to multifactorial mechanisms, and no single proposed theory alone can delay AD progression. Therefore, radical-scavenging activities (such as DPPH radical-scavenging activities), AChE inhibition, and anti-Aβ peptide aggregation, together with neuroprotection against the Aβ_25-35_ peptide- or hydrogen peroxide-induced cell death of recycling seeds and rinds were investigated. Previously, we reported that the agricultural waste of water caltrop hulls and isolated active compounds of the hydrolyzable tannins tellimagrandin-I and tellimagrandin-II inhibited AChE activities, anti-Aβ_1-42_ aggregation, and the anti-non-enzymatic glycation of AD-associated factors in vitro and enhanced learning behavior developments in scopolamine-induced ICR mice. These could be recycled as the starting materials of functional foods to improve cognitive dysfunctions [22]. It was reported that the seed oil of passion fruit contains high amounts of linoleic acid (C18:2) and oleic acid (C18:1), which could be applied in the food, pharmaceutical, and cosmetics industries [28,49,50]. The dried seeds in the present study can not only provide valuable passion fruit seed oil through hexane extractions (with a recovery of 18.61–21%), but de-oiled seed powders (S) can also act as ingredients to produce functional foods that undergo oil peroxidation because polyunsaturated fatty acids during storage.

This study began with successive extractions of de-oiled seeds and rinds (Figure 1), and each extract was focused on anti-AD-associated factors, including DPPH radical-scavenging activities, AChE-inhibitory activities, anti-Aβ_1-42_ peptide aggregations, and neuroprotective functions against H_2_O_2_-induced and Aβ_25-35_ peptide-induced neuronal cell death in vitro (Figure 2). It might be possible that the neuroprotective activities of the extracts of the seeds and rinds of passion fruit include elevating cell viability and recovering neurite lengths. The scavenging order of the DPPH radicals of the extracts was S-50EE > S-95EE >> rind-95EE > rind-HWE > S-HWE. Under the same assay system, the IC_50_ of the DPPH radical-scavenging activities of Trolox (the positive control) was 20.81 μg/mL (83.14 μM) [23]. S-95EE and S-50EE performed better than Trolox and acted as the most potent DPPH radical scavengers with an IC_50_ less than 10 μg/mL among these five extracts. The order of the anti-Aβ_1-42_ peptide aggregations of extracts from the seeds and rinds was S-50EE > S-95EE >> S-HWE > rind-95EE > rind-HWE. S-95EE and S-50EE acted as the most potent inhibitors of Aβ_1-42_ peptide aggregations in vitro among these five extracts. Two stilbene-related compounds, piceatannol and scirpusin B (a piceatannol dimer), were isolated from S-50EE and identified in S-95EE, S-50EE, and rind-HWE (Figure 4). They also showed AChE-inhibitory activities, anti-Aβ_1-42_ peptide aggregations, and neuroprotective functions against Aβ_25-35_ peptide-induced neuronal cell death in vitro (Figure 3). It was reported that trans-scirpusin B (IC_50_, 1.0 μM) and cis-scirpusin B (IC_50_, 1.3 μM) exhibit disaggregation activities toward pre-formed Aβ_1-42_ aggregates in vitro. These activities were shown to be about twofold higher than those of trans-piceatannol (IC_50_, 1.9 μM) and cis-piceatannol (IC_50_, 2.1 μM) [51], although piceatannol (IC_50_, 0.34 μM) was twofold more effective than scirpusin B (IC_50_, 0.63 μM) in enhancing anti-Aβ_1-42_ aggregations in the present study (Figure 3C). The higher disaggregation activities of scirpusin B than those of piceatannol in the pre-formed Aβ_1-42_ aggregates [51] might be a result of lower interaction energy in the docking poses (Figure 6B). It was found that the more stable complex of scirpusin B/Aβ_1-42_ in the regions of residues 11 to 22 had lower interaction energy (−35.6229 Kcal/mol) compared with the piceatannol/Aβ_1-42_ complex in the regions of residues 18 to 24 (−25.1388 Kcal/mol). Therefore, it seemed that scirpusin B interacted more easily with the docking site far from the core segments (residues 16 to 21) for Aβ aggregations [48] than piceatannol and had a greater ability to disentangle the pre-formed Aβ_1-42_ peptide aggregates, as mentioned above [51]. Sato et al. reported that piceatannol-enriched 35% ethanol extracts of passion fruit seeds and piceatannol showed neuroprotective activities against Aβ_1-42_-induced neuronal cell death in retinoic acid-induced differentiated SH-SY5Y cell models [52]. The present study demonstrated the neuroprotective functions of different extracts of the seeds and rinds of passion fruit against H_2_O_2_-induced and/or Aβ_25-35_ peptide-induced neuronal cell death in vitro (Figure 2), and both piceatannol and scirpusin B showed dose–effect (0.5 to 5.0 μM) relationship in neuroprotective functions against Aβ_25-35_ peptide-induced neuronal cell death in vitro (Figure 3). It was noted that the pretreatment with 50 μg/mL of S-95EE, S-50EE, S-HWE, rind-HWE, or rind-95EE recovered shortened neurite lengths in H_2_O_2_-treated SH-SY5Y cells (Appendix A).

The use of scopolamine to induce temporarily impaired memories in animal models was achieved by occupying muscarinic receptors, which blocked acetylcholine function to cease neurotransmission; applying AChE inhibitors restored the forgotten behaviors [9]. In this study, a two-day passive avoidance test was conducted to evaluate the learning behaviors, which included test compounds or drugs consolidating long-term emotional memory amplified by electric food shocks and effective communication between the amygdala and the hippocampus to achieve improved learning and memory [53]. A daily pretreatment for 7 days with de-oiled seed powders (S) and rind-HWE at a dose of 600 mg/kg or with S-95EE and S-50EE at a dose of 250 mg/kg improved learning behaviors in scopolamine-induced memory-impaired ICR mice (Figure 7A). A modified protocol of daily pretreatment for 18 days with de-oiled seed powders (S) at a dose of 400 mg/kg or with rind-HWE at doses of 250 and 400 mg/kg improved learning behaviors in scopolamine-induced memory-impaired ICR mice (Figure 7B). To identify the active compounds in de-oiled seed powders and S-95EE or S-50EE, the daily pretreatment for 7 days with piceatannol at a dose of 50 mg/kg or scirpusin B at a dose of 40 mg/kg improved learning behaviors in scopolamine-induced memory-impaired ICR mice (Figure 7C). As expected, the piceatannol-treated and scirpusin B-treated mice exhibited lower levels of AChE activities in brain extracts (Figure 7D) and had significant differences (*p* < 0.05) compared with the control (only the scopolamine-treated mice). It was noted that the step-through latency (s) of mice in the retention trial of the positive control of the donepezil-treated group, the piceatannol pre-treated group, and the scirpusin B-pre-treated group showed no significant difference (*p* > 0.05, Figure 7C), which meant the comparable effects of the donepezil treatment and the 7-day daily pretreatments with scirpusin B (40 mg/kg) or piceatannol (50 mg/kg) on improving learning and memory function will need further investigation. There have been few reports concerning the effects of scirpusin B on animal disease models. Piceatannol and scirpusin B have shown ex vivo vasorelaxation activities in blood vessel models [38] and cardiovascular-protective effects [54]. Using HPLC chromatograms to analyze the fingerprints of the extracts, piceatannol and scirpusin B were identified in S-95EE, S-50EE, and rind-HWE. The piceatannol and scirpusin B contents in S-95EE were 6.90% and 9.67%, respectively; the piceatannol and scirpusin B contents in S-50EE were 2.51% and 16.99%, respectively. The amounts of scirpusin B were higher than piceatannol in S-95EE and S-50EE in de-oiled seed extracts, in which the scirpusin B and piceatannol were recognized as the major compounds in the seed extracts. Based on the animal experiments in the present study, it was proposed that piceatannol and scirpusin B are the active compounds of the de-oiled seed powders and seed extracts of passion fruit and can be identified as indicator compounds to monitor ingredients from seed extracts when preparing functional foods to promote neuroprotection and delay the onset of cognitive dysfunction. Rind-HWE demonstrated benefits in improving learning behaviors, but the active compounds and indicator compounds were not identified in the present study. The *C*-glycosyl flavonoids of luteolin-6-*C*-glucoside (isoorientin, 196.3 μg/g), vicenin (98.6 μg/g), and isovitexin (27.6 μg/g) were identified in the rind flours, and pretreatments with 40 mg of rind flour per mouse daily for 16 days showed enough anti-inflammatory effects to maintain the intestinal barrier and reduce colon damage in dextran sodium sulfate-induced C57BL/6 mice [34]. The peel extracts of purple passion fruit contain the three major components cyanidin-3-*O*-glucoside, quercetin-3-*O*-glucoside, and edulilic acid, and the oral administration of anthocyanin-enriched peel extracts of purple passion fruit (150 mg/day) for four weeks in a randomized, placebo-controlled, double-blind trial relieved symptoms in patients with asthma [55]. The peel extracts of purple passion fruit at doses of 50 and 200 mg/kg have shown antihypertensive activities in spontaneously hypertensive rat models [35]. Piceatannol (0.34%) and scirpusin B (0.51%) were also identified in rind-HWE in lower amounts, and these anthocyanin-related compounds may be major and active compounds in rind-HWE and will need further investigation. The effective doses used in animal experiments could be mutually translated via the normalization of the body surface area to those used in subject trials by multiplying the Km factor (body weight (kg) divided by body surface area (m^2^)) and dose usages (mg/kg) as follows: the equivalent human dose (mg/kg) × Km factor (human) = animal dose (mg/kg) × Km factor (experimental animal) [56]. Therefore, the human concentration equivalent of a 7-day treatment with de-oiled seeds and rind-HWE (600 mg/kg) for scopolamine-induced ICR mice was calculated to be 48.65 mg/kg, and an adult weighing 60 kg is expected to receive similar beneficial effects by taking about 2.92 g per day; the human concentration equivalent of a 7-day treatment with S-95EE and S-50EE (250 mg/kg) for scopolamine-induced ICR mice was calculated to be 20.27 mg/kg, and an adult weighing 60 kg is expected to receive similar beneficial effects by taking about 1.22 g per day; the human concentration equivalent of an 18-day treatment with de-oiled seeds (400 mg/kg) for scopolamine-induced ICR mice was calculated to be 32.43 mg/kg, and an adult weighing 60 kg is expected to receive similar beneficial effects by taking about 1.95 g per day; the human concentration equivalent of an 18-day treatment with rind-HWE (250 mg/kg) for scopolamine-induced ICR mice was calculated to be 20.27 mg/kg, and an adult weighing 60 kg is expected to receive similar beneficial effects by taking about 1.22 g per day. However, further investigations will be needed in subject trials. It has been reported that piceatannol-3′-*O*-β-D-glucopyranoside can penetrate and cross the blood–brain barrier [57], and piceatannol has shown protective capacities related to brain microvascular endothelial cells in oxidative and inflammatory cell models [58]. No pharmacokinetic properties of scirpusin B have been reported. There are limitations in the present animal experiments. First, no information was reported on whether the active compounds, such as piceatannol and scirpusin B, could penetrate the blood–brain barrier and serum concentrations; second, the scopolamine-induced ICR mouse models based on the cholinergic hypothesis did not fully cover the pathologies of AD; third, there was a loss of rind powders used in the present animal experiments, and no information was provided concerning the active compounds of the rind extracts used in the animal experiments; fourth, galactose-induced oxidative stress aging mouse models [59] could be used to evaluate the effects of the seeds and rinds of passion fruits on normal aging; and fifth, for the impact of using the seeds and rinds of passion fruits to improve cognitive dysfunction, experimental animal numbers could be increased in the future experiments.

## 5. Conclusions

The recycled seeds and rinds of passion fruits from juice-processing factories may be sustainable utilities for the environment and may also be good resources for recovering ingredients for the development of functional foods for unmet medical needs in delaying the onset of cognitive dysfunction.

## Data Availability

All figures and data used to support this study are included in this article.

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
