# Peer review of "Extracts and Scirpusin B from Recycled Seeds and Rinds of Passion Fruits (Passiflora edulis var. Tainung No. 1) Exhibit Improved Functions in Scopolamine-Induced Impaired-Memory ICR Mice"

_antioxidants, 2023, doi:10.3390/antiox12122058_

Round 1
Reviewer 1 Report
Comments and Suggestions for Authors
This paper is an extremely well-written paper about the potential use of ‘recycled seeds and rinds of passion fruits’ for the treatment of AD or impaired cognition. If ‘recycled seeds and rinds of passion fruits’ were actually effective for the amelioration of AD, it would be really wonderful. What is more, if what was once regarded as just the ‘waste’ were re-used and recycled in such a wonderful way, it would be all the more wonderful because it would quench the environmental problems associated with what to do with that waste. This paper is really commendable. I have a few minor comments about the paper.
#Results
-Figure 2 and 3, ‘blank and control’: I guess ‘blank’ is when just DMSO is applied, and ‘control’ is when DMSO and hydrogen peroxide/amyloid-beta peptide are applied, right? At first glance, it was difficult to understand what these ‘blank’ and ‘control’ exactly are. The exact explanation of ‘blank’ and ‘control’ should be given in the section, for example, in the section ‘2.4.4.’, which introduces these terms at present just as ‘0.1% DMSO (the control and the blank)’ in line 322.
-This comment, which is about Figure 7, is similar to the above. I guess ‘blank’ is when just distilled water and PBS are applied, and ‘control’ is when distilled water and scopolamine are applied, right? I want the exact explanation of ‘blank’ and ‘control’ somewhere in text if it is now nowhere in the manuscript.
-Figure 7E and 7F: Here, lowercase a, or uppercase A, B, AB, or C is given. In the legend, it says these lowercase or uppercase alphabet letters indicate significance (P < 0.05). But it was difficult to understand why these different As, Bs, Cs, or whatsoever had to come here. For example, an A happens to mean ‘it is significantly (P < 0.05) different from blank’??? In the legend of this figure, an explanation of what As, Bs, Cs, or whatsoever means, which is helpful for the readers, should be written.
#Abstract or Other parts of the manuscript (when applicable)
For example, in the last line of the Abstract, it says, ‘neuroprotection and delaying the onset of neurodegerative disorders’. This is also true for the last line of the Introduction. But this is incorrect. This paper or the experiments of this paper have been devoted to, like, ‘memory impairment, cognitive dysfunction, or AD’, but not to ‘neurodegerative disorders’. If we use the term ‘neurodegerative disorders’, it could include other diseases, like MSA, PSP, or whatsoever. So the authors should not use the term ‘neurodegerative disorders’ and restrict their use to, like, ‘memory impairment, cognitive dysfunction, or AD’, in the parts where this is applicable.
#As is well-known, the pathologic basis of AD is the aging of the brain. Cognitive impairment can also happen in normal aging. I want to know the future applicability of ‘recycled seeds and rinds of passion fruits’ to such normally (non-pathologically) aged people in the Discussion. Just a few comments are okay.
#A minor grammatical editing is advisable.
Ex
-2.6., line 350: models were following > models were conducted following (?)
-2.6. line 352: which the approval number > of which the approval number OR whose approval number (?)
-2.8., line 438-439: The …(ANOVA) and…Tukey’s test was > The …(ANOVA) and…Tukey’s test were (?)
-Results, line 450: investigated the effects on AD-associated factors > investigated for the effects of AD-associated factors
-etc.
Comments on the Quality of English LanguageFairly good.
Reviewer 2 Report
Comments and Suggestions for Authors
In the present manuscript, the authors investigated the beneficial effects of various extracts from recycles seed and rinds of Passion Fruits in both in vivo and in vitro models. This manuscript did very relevant research, which is of significance to the field of Alzheimer’s disease, but failed to emphasize the relevance and to present the results. Here are the comments from the reviewer.
Major comments:
1. This manuscript is well-written, but too long and contains lots of irrelevant information, especially in the introduction and results part, which must be condensed.
2. The language editing is required.
3. Study design:
For the in vitro study:
1) DPPH and ACHE are related mechanism of AD. But why these two parameters were selected for evaluating the benefits of the extracts in the in vitro study? Please explain the rationale and re-organize the presentation of the results.
2) In the cell study, the concentration range was very narrow and there was little difference between each concentration, which made these concentrations less representative. Please explain why these concentrations were selected for each extract.
3) Hydrogen peroxide was applied to cells for impairment and DPPH radical scavenging activity was measured in this study. Therefore, the reviewer is expecting furthermore anti-oxidant evidence for the beneficial effects of these extracts.
For the animal study:
1) Did the present study obtain any ethical approval?
2) There were only 5 mice in each group, which appears to be too little and may cause low replicability and lack of reproducibility in the results. Please provide the calculation in details to justify how the present sample size was determined.
3) Single dose for each extract was applied in the animal treatment. Please justify why the present dosage(s) were selected.
4) How long did the animal treatment last for?
5) The authors are strongly recommended to show the flow of animal experiment by using a diagram.
4. Results:
1) The results were too long. Please revise to make it brief and precise.
2) Figure 2D and 2E contain too much groups and are very hard to read. Please separate.
3) The results were not logically presented. Please re-organize.
5. Discussion was too long and contained lots of irrelevant information. Please re-write.
Comments on the Quality of English LanguageLanguage editing is required.
Round 2
Reviewer 2 Report
Comments and Suggestions for Authors
I have no further comments regarding the study design. But the manuscript, especially the introduction and results, are too long and must be narrowed down!
Comments on the Quality of English LanguageEnglish is fine.
Author Response
Responses to Reviewer 2
- I have no further comments regarding the study design. But the manuscript, especially the introduction and results, are too long and must be narrowed down!
Ans: Thanks for reviewer’s suggestion. The revised MS was more concise by editing the result part and introduction part. Finally, the 22 pages containing 12937 words were narrowed down to 21 pages with 12201 words in total.